# The Cost of Downstream Adverse Outcomes Associated with Allogeneic Blood Transfusion: A Retrospective Observational Cohort Study

**DOI:** 10.3390/healthcare13050503

**Published:** 2025-02-26

**Authors:** Michelle Roets, David John Sturgess, Kerstin Hildegard Wyssusek, Sung Min Lee, Melinda Margaret Dean, Andre van Zundert

**Affiliations:** 1Department of Anaesthesia, Royal Brisbane and Women’s Hospital, Faculty of Medicine, University of Queensland, Brisbane, QLD 4006, Australia; kerstin.wyssusek@health.qld.gov.au (K.H.W.); sungmin.lee@uq.net.au (S.M.L.); a.vanzundert@uq.edu.au (A.v.Z.); 2Princess Alexandra Hospital, Faculty of Medicine, University of Queensland, Brisbane, QLD 4102, Australia; d.sturgess@uq.edu.au; 3School of Health and Behavioral Sciences, University of the Sunshine Coast, Sunshine Coast, QLD 4556, Australia; mdean@usc.edu.au

**Keywords:** allogeneic blood transfusion, cost of adverse outcomes, intraoperative cell salvage

## Abstract

**Background:** ‘Downstream’ adverse outcomes associated with transfusion-related immune modulation (TRIM) occur postoperatively. The potential associations between these outcomes (and costs) and perioperative transfusion are often not considered by clinicians and therefore underestimated. When considering TRIM, many advantages of intraoperative cell salvage (ICS) were previously confirmed. **Methods:** The main aim of this retrospective observational study was to evaluate the cost implications associated with perioperative adverse outcomes following allogeneic blood transfusion (ABT). Secondly, further analysis considered downstream costs following ICS. This manuscript does not aim to provide evidence of improved outcomes following ICS compared to ABT. These outcomes were previously demonstrated. Instead, it is important to consider downstream cost implications if patients receive ABT, despite previously proven benefits related to ICS. Surgical patients (n = 2129) receiving blood transfusion at the Royal Brisbane and Women’s Hospital (Queensland, Australia) (2016–2018) were included: receiving ICS only (n = 115), allogeneic red blood cells (RBCs) only (n = 1944), or RBCs and ICS (n = 70). Data retrieved from eight hospital databases were exported, and a novel Structured Query Language (SQL) database was developed to link data points. Adverse outcomes previously associated with TRIM were assessed using International Classification of Diseases-10 (ICD-10) coded data. Generalised linear models were used to model costs and adjust for confounding factors. **Results:** Most adverse outcomes (≥3) occurred following RBCs and ICS (37.1%), followed by RBCs (23.7%) and ICS (16.5%). As potentially important determinants of overall expenditure, the lowest marginal mean intensive care stay (days, cost) was after ICS (2.1 days, AUD 10,027), followed by RBCs and ICS (3.8 days, AUD 18,089), and then RBCs (5.5 days, AUD 26,071). When considering blood products (other than packed red blood cells), the average cost per patient was lowest for ICS (AUD 48), followed by RBCs (AUD 533) and RBCs and ICS (AUD 819). **Conclusions:** We confirmed that the cost associated with allogeneic blood transfusion was significant; patients receiving packed red blood cells (pRBCs) experienced more adverse outcomes and higher hospital costs than those receiving ICS. These results are limited to retrospective data and require further prospective validation.

## 1. Introduction

Blood transfusion saves lives. However, despite improved transfusion techniques and practices developed over centuries [1], adverse outcomes still occur [2,3]. These outcomes include postoperative infection [4], respiratory failure [5], myocardial infarction, stroke [6], renal failure, multiorgan failure [7], thromboembolism [8], other immunological consequences [9], traditional transfusion complications [10], and transfusion complications in patients who regularly use anticoagulation medication. Even though often underestimated, the costs associated with these adverse outcomes are significant [6]. Downstream adverse outcomes related to transfusion and immunological consequences occur at a later stage within the perioperative journey in the presence of many other confounding factors. It is therefore difficult to identify or attribute these adverse outcomes to a specific transfusion event [6].

According to the NBA (National Blood Authority, Australian government), the overall cost of ABT in Australia (population 25.7 million) [11] is over AUD 1 billion per year [12]. This does not include costs related to the treatment of adverse outcomes. Patient blood management (PBM) strategies reduce blood product requirements, adverse outcomes, and related costs. Intraoperative cell salvage (ICS), as a PBM strategy, is a safe alternative to ABT and permits autologous (own) blood lost during surgery to be collected, processed, and reinfused [13,14,15]. We know that ICS provides immunological benefits, improves outcomes, and reduces blood product requirements, as was previously confirmed during clinical studies and in vitro [4,14,15]. Despite this evidence, many patients who currently receive ABT do not have the opportunity to benefit from ICS.

The analysis of transfusion costs for both ICS and ABT should ideally include all aspects related to direct costs, indirect costs, and adverse outcomes to capture the full cost burden [6,9,16,17]. The requirement for specialised staff and equipment led to ICS being perceived as costly to healthcare services. Over the past three decades, most studies have therefore focused on comparing costs related to consumables, device purchase, and maintenance for ICS with the purchase cost of allogeneic packed red blood cell (pRBC) units and have not included costs related to adverse outcomes [18,19].

Outcomes related to TRIM are extensively studied in the literature (PHD thesis) [2,14,20]. Modern science considers outcomes beyond the traditionally studied outcomes (i.e., infection and cancer recurrence) [15,21]. We could, however, not find any studies that included all transfusion-related adverse outcomes associated with immune modulation (TRIM) (e.g., not only wound infection but also pneumonia, renal failure, cardiovascular and stroke-related outcomes, etc.), used objective and standardised ICD-10 coding, and included an assessment of the significance and impact of the associated cost burden (considering pRBCs and other blood products, length of stay in hospital (LOS), and length of stay in the intensive care unit (ICU LOS)). We agree with Ning et al., who in a narrative review considers the value and challenges relevant to “big data” in transfusion medicine in the future. There is a potential that database analysis and artificial intelligence will, in the future, lead to improved clarity of the relevance of outcomes related to TRIM and associated organ dysfunction [22]. Firstly, this novel study was designed to satisfy these existing knowledge gaps identified through an extensive literature review. Secondly, study methods (with targeted data collection) were designed to overcome some limitations and areas of controversy identified in previous studies through the inclusion of a large sample of transfused patients (n > 2129), all procedures and subspecialties currently requiring transfusion, and three clinically relevant study groups, all while considering multiple adverse outcomes associated with TRIM.

From a previous audit at the Royal Brisbane and Women’s Hospital (RBWH), we identified that many patients receiving RBCs could benefit from ICS instead (Roets et al. [23]). According to a staff survey considering “experiences, conceptions and barriers” to the implementation of ICS, the availability of additional trained staff and costs were major obstacles to the wider use of ICS (Tognolini et al. [24]). This study assessed whether costs related to adverse outcomes were different when considering transfusion models commonly represented in clinical care (i.e., ICS, RBCs, or RBCs and ICS).

The novelty of our study therefore lies in the study design (carefully considering each step in a pragmatic way to overcome familiar challenges), which was conducted in real life, within best practice-guided perioperative care, across a wide field of surgical subspecialties and procedures, in a large sample size (n = 2129), and considering objective ICD-10 coded adverse outcomes as determinants of overall expenditure (Appendix A).

## 2. Materials and Methods

### 2.1. Study Design and Participants

The RBWH is a quaternary referral teaching hospital, the largest provider of healthcare services in Queensland, Australia, and provides care for patients undergoing an estimated 33,000 surgical procedures each year (including all patients > 12 years of age, excluding cardiac surgery). The RBWH was an early adopter of ICS technology and one of the first to implement an in-house ICS service in Australia in 1992. ICS is currently standard practice for many surgical procedures where major blood loss is expected. ICS is booked at the RBWH at the discretion of the surgeon and anaesthetist when major blood loss (i.e., more than 500 mL) is expected, for any procedure. ICS is available within and outside office hours, for elective and emergency cases, every day of the year at the RBWH (and not only for a list of predefined procedures). There are no absolute contraindications to ICS [14]. This retrospective study quantified patients undergoing elective and emergency surgical procedures at the RBWH (n = 2129, 2016–2018) who received one of three modes of transfusion: ICS only (ICS), allogeneic pRBCs only (RBC), or allogeneic pRBCs and ICS (RBCs&ICS). Administrative data were used, with an ethics exemption from the RBWH Human Research Ethics Committee (HREC/16/QRBW/437).

### 2.2. Patient and Public Involvement

Patients and/or the public were not involved in the design or conduct of this study.

### 2.3. Data Collection

Data retrieved from eight hospital databases were exported, and a novel structured SQL database was developed to link data points (Table 1).

Only clinical variables and outcomes relevant to the specific surgical procedure (i.e., 24 h before, during, and immediately after) and the related transfusion episode were assessed. An SQL database was developed, using exported data, to link the data points. Databases were not linked due to compatibility challenges between software systems used in relevant databases. Instead, data retrieved from eight hospital databases were exported into Excel and then imported into a Structured Query Language (SQL) database to link data points, with a query analysis in three monthly cycles over a 2-year period. Specific strategies were followed at each data import and linkage step (i.e., collect, connect, and clean) involving information technology experts. In addition, each data import step was checked by five study members against the original exported data. An additional check (data cleaning) was included within the statistical analysis. Exporting data from existing hospital databases rather than using manually imported data did minimise data entry errors. Extra checks were included for doubled data and outliers. Data collected included sociodemographic data (patient age, body mass index (BMI), and sex) and clinical variables (surgical procedure, subspecialty, and patient comorbidities). Outcomes of interest included LOS, ICU LOS, ICD-10 coded adverse outcomes, transfused allogeneic blood products (e.g., pRBC units, pools of platelets), and ICS. We did not consider traditional clinical “outcomes”, as these have been extensively studied previously (i.e., volume blood lost, traditional transfusion thresholds, etc.). We specifically considered variables that relate to immune consequences and have cost implications for health services [22,25]. Each variable measured (called “outcomes” in this manuscript) was individually chosen specifically because they are considered to be associated with the pathophysiology identified within the TRIM literature (i.e., the immunological consequences of transfusion) and incur cost from a healthcare point of view. The data collection included the setup, processing, and transfusion of ICS, but only patients who received ICS transfusion were included for the purposes of the statistical analysis. The decision to transfuse or discard processed ICS blood was made by the treating team on clinical grounds and considering current best practice in this field.

### 2.4. Comorbidity Categories

In addition to immune modulation associated with transfusion, patient comorbidities influence surgical outcomes (Table 1). We adjusted for patient comorbidities, and ongoing disease was considered within organ-specific categories, including heart disease (e.g., ischaemic heart disease, cardiomyopathy, pulmonary hypertension, valvular disease, and arrhythmias), cerebrovascular incidents (e.g., transient ischaemic attack (TIA)), diabetes mellitus (DM) (e.g., DM type I and II and gestational diabetes), renal disease (e.g., acute and chronic renal impairment), hypertension (e.g., essential and preeclampsia), hypercholesterolaemia and respiratory disease (e.g., asthma and chronic obstructive airway disease), and cancer (unless resected or in remission >2 years, i.e., leukaemia). Haematological comorbidities included all diseases of haematological origin (e.g., idiopathic thrombocytopenic purpura, hereditary haemorrhagic telangiectasia, thalassemia, and thrombocythemia). The number of comorbidities per patient was counted, and patients were categorised as those with 0, 1, 2, or 3 or more comorbidities.

### 2.5. Adverse Outcome Categories

In total, 171 ICD-10 coded adverse outcomes, relevant to the specific surgical procedure and formally coded by professional coding staff, were extracted from the HIS (Health Information System) database. The total number of adverse outcomes and the number of adverse outcomes for each case within organ-specific categories were recorded (Table 1). To enable comparisons across groups, patients were categorised as those with 0, 1, 2, or 3 or more adverse outcomes. The treatment of anaemia holds cost implications for healthcare providers [25,26]. In our study, all ICD-10 codes with cost implications relevant in the management of patients with anaemia (i.e., D62, D50, D51, D53, D64, etc.) [27] and all causes of preoperative anaemia and postoperative anaemia (both associated with a higher transfusion incidence) were considered within the anaemia category for the purposes of the analysis [28].

### 2.6. Study Outcomes Assessed

As potentially important determinants of overall expenditure, LOS, ICU LOS, associated ICD-10 coded adverse outcomes, blood product requirements, and related costs were compared across the three study groups [6]. Previously defined aggregated costs for LOS (AUD 2140.81 per day) [29] and ICU LOS (AUD 4717.32 per day) [30] were applied. The direct costs of blood products (as defined by the National Blood Authority in Australia) [12] were used to evaluate blood product cost. All cost values were adjusted for inflation to 2020 in Australian dollars (AUD).

### 2.7. Statistical Analysis

Statistical analyses were performed using R version 4.1.1 (R Core Team, 2018, Vienna, Austria). For descriptive analysis, continuous data were expressed as the means and standard deviations (SDs) for symmetrically distributed variables and medians and interquartile ranges (IQRs) for nonsymmetrically distributed variables. Categorical variables were summarised as frequencies and percentages. Chi-square or Fisher’s exact tests were performed as appropriate to explore associations between categorical variables. Kruskal-Wallis tests were used to compare continuous non-normally distributed variables across the groups. Where there were large numbers of zeros, k-sample Anderson Darling tests were used (i.e., comparing blood products across treatment groups). Hospital LOS was modelled using a generalised linear model (GLM) with a gamma family of distributions and log link function. ICU LOS and ICU LOS cost were modelled using hurdle models (two-part models) where the occurrence of an ICU admission was modelled with logistic regression, and the ICU LOS (number of days and related costs) was modelled using generalised linear models with a gamma distribution and log link function. The study design and statistical methods were chosen following an extensive consultation with an experienced biostatistician and health economist from the start of the study to manuscript preparation. A generalised linear model with a gamma distribution and log link (commonly used to model costs and length of stay in healthcare data) was appropriate to be used when the outcome was positive, continuous, and skewed. The gamma distribution could accommodate differing levels of skewness and provided a reasonably good fit when there was heteroscedasticity. The log link function allowed for the modelling of multiplicative relationships (i.e., evaluate relative changes). It also reduced the impact of extreme values and stabilised variance. All the statistical models presented were run with the treatment group alone as well as with adjustment for the potential known confounding factors of age, sex, BMI, pre-existing comorbidities, and adverse outcomes. *p* < 0.05 was considered to indicate a statistically significant difference.

## 3. Results

The total number of patients included in the statistical analysis was 2129 (across all surgical subspecialties) (Table 2).

Most patients received only RBCs (n = 1944). Even though ICS was booked and collected for 234 patients, 185 received ICS transfusion and were included in both the ICS (n = 115) and RBCs&ICS (n = 70) groups.

The median age at the time of surgery was 58 years (IQR 38–71), and the median BMI was 26.8 kg/m^2^ (IQR 23.1 to 31.6) (Table 3), with statistically significant differences across groups for sex. The numbers of comorbidities were similar. Most adverse outcomes (≥3) occurred after RBCs&ICS (37.1%), followed by RBCs (23.7%) and ICS (16.5%) (*p* < 0.001). ICS represented the largest proportion with no adverse outcomes (36.5%).

The marginal mean LOS (and cost), unadjusted and adjusted, was not significantly different across the study groups; it was similar considering age and BMI, significantly higher for males, and increased with increasing numbers of comorbidities and adverse outcomes (Table 4). The marginal mean LOS cost (95% CI) was AUD 31,803 (25,423–39,784) for patients with no adverse outcomes and AUD 48,752 (40,686–58,418) for patients with ≥3 adverse outcomes.

ICS was associated with a reduction in ICU LOS (days and cost) before and after adjustment (Table 5). Although statistically significant, the ratio of change of 1.0 ICU LOS for each year of age was not clinically significant (95% CI 1.0–1.0). ICU LOS did not change significantly with BMI. The marginal mean ICU LOS (and cost) was similar for those with increasing numbers of comorbidities but increased with increasing numbers of adverse outcomes.

The overall cost for blood products was AUS 4,791,979 (Table 6). To contextualise the relevant total product costs for each group, it was AUD 5599 for ICS, AUD 201,471 for RBCs&ICS, and AUD 4,584,909 for RBCs. The distributions of the numbers of units transfused and related cost (on a per patient basis) of pooled platelets, plasma, cryoprecipitate, and apheresis cryoprecipitate were significantly different across groups; they were lowest for ICS, followed by RBCs, and then RBCs&ICS. When considering the average total cost (and when excluding costs for packed RBC units), the cost of other blood products for ICS was the lowest, followed by RBCs&ICS and RBCs.

When considering organ-specific categories, 2129 patients experienced 4029 adverse outcomes, and 89.9% experienced at least one adverse outcome (Table 7). When considering study groups, adverse outcomes occurred more commonly for RBCs (1777/1944, 91.4%) and for RBCs&ICS (64/70, 91.4%) and less frequently for ICS (73/115, 63.5%). Some patients experienced more than one adverse outcome within one category and/or in more than one category. Of the 2129 patients in our study, 82 died in the hospital (3.9%).

## 4. Discussion

A detailed retrospective study to assess the impact of perioperative transfusion on adverse outcome-related costs in a large quaternary referral hospital was conducted. Costs related to blood product requirements and ICU LOS were lower following ICS than following RBCs or RBCs&ICS. ICS was associated with the lowest frequency of adverse outcomes overall. Despite the previously reported results, ICS was not associated with a significantly reduced LOS.

When designing this study, the aim was to overcome some limitations identified by the authors of previous studies (i.e., the reasons why previous studies have not sufficiently addressed cost implications). Studies considering ICS costs varied from simple consumable usage calculations to detailed and extensive cost-effectiveness analyses [14,18,31,32], which often included small sample sizes, targeted specific surgical procedures commonly considered within the traditional ICS literature [33], and excluded other procedures associated with major blood loss and those with additional surgical complexity (e.g., repeat procedures). Despite the potential clinical value often associated with ICS during these procedures, they were often excluded because they added complexity to the study designs [34]. Many described ICS as “cost-effective” when its use resulted in a reduced (units of) RBC requirement (i.e., the cost of units of RBCs avoided) or an overall reduction in blood loss [35] and did not consider costs associated with other blood products, overhead costs, and adverse outcomes [14,18,19,36]. ICS is not standardised across countries and hospitals, with different equipment, resource availability, techniques (i.e., washed and unwashed ICS), indications defined by local clinical policies, and different currencies across hospitals and countries [14,18]. Individual institutional considerations differ significantly (i.e., case acuity, procedure type, duration of surgery, inclusion or exclusion of certain cases with perceived cancer or infection risks, blood loss volume, relevant case numbers, available expertise, and additional staffing and training requirements) [32]. Future multicentre international trials that consider these potential benefits are needed. However, the design of such studies that will overcome the challenges related to institutional differences in the ICS procedure will be complicated. In most large centres, fewer patients receive ICS compared to those receiving RBCs (due to availability of service), often resulting in studies considering small sample sizes for ICS. Smaller studies may not be adequately powered to assess specific outcomes that occur infrequently (e.g., wound infection) [14], while the incidence of all types of infection (TRIM) coded in this large study population was 23.9% (509 outcomes/2129 patients). Davies et al. performed a cost-effectiveness analysis of ICS and recommended future observational studies with large sample sizes to overcome some of these limitations [18]. The presence of clinically relevant confounding factors in transfusion research and the inability to truly blind an autotransfusion device during randomised controlled trials (RCTs) are additional complicating factors. Despite these limitations, scientific evidence demonstrates the potential cost benefits. Costs associated with reductions in allogeneic blood product requirements and adverse outcomes [4,37] may offset the equipment and staffing costs associated with ICS [14,38].

In a ground-breaking study by Leahy et al., the implementation of a PBM programme mainly focused on the diagnosis and treatment of preoperative anaemia [6]. Significant reductions in ABT and subsequent cost savings when considering saved blood products, LOS, emergency readmissions, and activity-based costs of transfusion were confirmed. ICS may reduce costs in a similar way [14,18]. To ensure robust evidence that will inform a business case for change, costs to healthcare services associated with adverse outcomes (TRIM) (i.e., considering investigation, treatment, LOS, and ICU LOS) should be considered (6, 23, 1, 10). While the obvious costs associated with ICS, to purchase equipment and consumables, implementation, education, and additional staffing requirements, have been thoroughly investigated [15,18,32,37], this study now provides evidence of the costs associated with adverse outcomes [4,5,7,8,39,40,41], potentially avoided by ICS, and previously unknown.

The authors agree with Mukhtar et al. (2013), who described the challenges involved in enabling data collection within PBM projects during a government-funded project [42]. In this study, it was not possible to directly link data across existing databases before data export (i.e., to ensure that measured outcomes were directly associated with a specific surgical event). Databases were not linked due to compatibility issues and different software languages between software systems used in the relevant databases. Instead, data were retrieved from eight hospital databases, exported into an Excel spreadsheet, and then imported into a specifically designed novel Structured Query Language (SQL) database, developed to link the relevant data points. The import of data within three monthly cycles over a 2-year period allowed for extensive query analysis to ensure that potential errors were mitigated. Specific strategies were followed at each data import and linkage step (i.e., collect, connect, and clean) involving professional information technology experts. In addition, each data import step was checked by five study members against the original exported data. An additional check (data cleaning) was included within the statistical analysis. Exporting data from existing hospital databases rather than using the manual importing of data did minimise data entry errors. Extra checks were included for doubled data and outliers. Traditionally, hospitals collected data to assess the number of specific blood products transfused per admission period, units wasted, and direct transfusion-related events (for example, ABO incompatibility). These requirements are now changing. To ensure that modern PBM requirements are met in the future [9,32], additional data points that allow for the assessment of costs associated with clinical outcomes will require additional sophisticated data coding expertise and software solutions, including relevant funding allocation [43,44]. Even though an assessment of individual costs related to each adverse outcome would be ideal, this was not logistically possible. Instead, investigators evaluated the costs associated with LOS, ICU LOS, and blood products transfused while considering differences in adverse outcomes for the same patients across study groups.

The highest number of cases requiring transfusion were from the subspecialities of orthopaedics (n = 468), obstetrics and gynaecology (n = 427), general (n = 363), vascular (n = 284), and gastrointestinal surgery (n = 111) (Table 2). The safety of ICS has improved in recent decades. Traditional contraindications, previously excluding its use for many surgical procedures, are no longer relevant. Even though these study groups were not significantly different when considering age, BMI, and comorbidities, more ICS patients were male (Table 3). While the ICS technique is standardised at the RBWH, indications for its use are not. During procedures where infection risk and potential cancer cell contamination warrant caution, clinicians evaluate the risk–benefit balance for each case before booking ICS according to international standard best practice [13]. For example, ICS is commonly used during radical prostatectomy for cancer at the RBWH. The investigators considered these implications within a detailed literature review and recently published as a book chapter (15 July 2022): “Roets M, Tognolini A, Dean M. Can We Overcome the Obstacles to Modern Intraoperative Cell Salvage Transfusion? A Detailed Review of Current Evidence” [45]. For the purposes of this study, the investigators therefore included all patients receiving blood transfusion, independent of cancer and infection risks. The investigators did not assess surgical duration, estimated blood loss, or haemoglobin level (“transfusion triggers”) as outcome measures. The decision to transfuse (based on best international standard practice) currently includes these traditional measures and many other factors, transfusion triggers, ongoing blood loss, the potential for postoperative blood loss, presence of infection, presence of antiplatelet medications, patient-specific cardiovascular requirements, etc. Instead, to ensure that all these factors were considered, this study reports the clinical transfusion requirement (i.e., was transfusion required yes/no), which includes all the abovementioned factors at the time of surgery, and the volume of blood products transfused (numbers of units for ABT and volume (ml) for ICS) as outcome measures. Furthermore, this study did not only consider the cost of RBC units but also the cost of other blood products (different from traditional ICS cost studies). No patients who received RBCs while undergoing gastrointestinal, plastic, ear, nose and throat, maxillo-facial, radiation oncology, thoracic, or eye surgery received ICS. These procedures represent potential opportunities to increase ICS use. It is worth noting that those who receive transfusion extend beyond the traditional vascular and orthopaedic surgery subspecialties and therefore represent additional ICS opportunities. Not surprisingly, those who started their surgical journey with more pre-existing comorbidities and were relatively more unwell therefore experienced longer hospital stays (although this was similar across study groups). The adjusted cost difference associated with LOS between those with none (AUD 34,567) and those with ≥3 comorbidities (AUD 49,514) was AUD 14,947. The association between ABT, adverse outcomes, and LOS, despite confounding factors, was previously confirmed [6,14,41]. The mean LOS was significantly longer for males, with an associated LOS cost difference of AUD 12,390. Even though LOS (and related cost) was lower following ICS, this difference was not statistically significant (unadjusted and adjusted) (Table 4). There was, however, an increasingly significant difference in the marginal mean LOS (days, cost) for those who experienced more adverse outcomes: 7.9 days between those with none (14.9, AUD 31,803) and those with ≥3 adverse outcomes (22.8, Aud 48,752).

The potential association with significantly reduced ICU LOS following ICS was confirmed. Following adjustment, the marginal mean ICU LOS cost for ICS (AUD 10,027) represented a significant saving compared to RBCs&ICS (AUD 18,089) and RBCs (AUD 26,071). Furthermore, age, BMI, and male sex were significantly associated with increased ICU LOS, but comorbidities showed no statistical evidence of an association. ICU LOS also significantly increased for those who experienced two or more adverse outcomes. The relevant cost implications confirmed were AUD 10,400 if no adverse outcomes occurred and AUD 22,285 if ≥3 adverse outcomes occurred.

A comparison of costs related to blood product requirements is essential, considering the annual blood product cost in Australia of AUD 1.196 billion [12]. Our study confirmed an overall perioperative blood product cost of AUD 4,791,979 including RBCs (AUD 1,100,913 excluding RBCs). Potential savings were identified following ICS, with significant reductions in blood product requirements (i.e., RBCs, pooled platelets, fresh frozen plasma, and cryoprecipitate). When considering blood products other than RBCs per patient, the ICS group had the lowest average cost (AUD 48), followed by the RBCs group (AUD 533) and then the RBCs&ICS group (AUD 819). The potential to reduce RBC requirements by using ICS instead (for specific procedures) was previously confirmed [14]. The authors of this manuscript encourage future study into the potential association between ICS and reduced blood products (other than RBCs). This study provides evidence that considers the relevant size of the related cost burden across all transfused patients.

The treatment of adverse outcomes significantly rises the cost of healthcare for patients and healthcare providers [18,46,47,48]. The proportion of patients with adverse outcomes was significantly lower following ICS at 63.5% compared to both RBCs and RBCs&ICS at 91.4% (Table 7). A similar association was also reflected in the relatively increased LOS and ICU LOS when adjusting for adverse outcomes. Although not statistically significant, the differences in this trend across study groups may suggest a lower incidence of infection, similar to previous studies [4]. When comparing adverse outcomes within organ-specific subcategories across groups, statistically significant differences were identified for respiratory-, cardiovascular-, and anaemia-related outcomes. Interestingly, ICS was not associated with reduced respiratory adverse outcomes; lowest in the RBCs group (20%), followed by ICS (25.2%), and then RBCs&ICS (31.4%). From a cardiovascular point of view, ICS seemed protective, as adverse outcomes were lowest in the ICS group (38.3%) compared to the RBCs (48.0%) and RBCs&ICS groups (65.7%). Renal-, cerebrovascular-, medication-, and thromboembolism-related events in our study were uncommon, and differences were not statistically significant across study groups. However, our sample size of 2129 may be too small to find significant effects in rare events (for example mortality), as previously reported following ABT [49]. Future ICS research should not only concentrate on historical equipment and staffing costs but also consider costs avoided when adverse outcomes are prevented. Current evidence and expert opinion should guide the allocation of funding in these circumstances. ICS would never be relevant for all surgical procedures, but resource allocation toward its wider implementation would be justifiable.

There were some limitations to our study, including its retrospective data collection, single-centre design, and observational nature. Many aspects specific to ICS research considered in our study cannot be assessed through traditional multicentre RCT designs. Davies et al. (2006), after a cost-effectiveness analysis considering ICS, concluded that observational studies are required to answer some of these questions [18]. By using observational data, this study was able to evaluate an unaltered patient cohort, where the same patients, at the same time, received the same clinical level of care, the same local indications for transfusion, and ICS procedures were standardised across all the study patients. The adverse outcomes considered in this study occur commonly today during real-life healthcare. Clinical admission criteria (surgical or anaesthetic) dictate ICU admission at the RBWH and include factors such as extreme age ranges, BMIs, and those with multiple comorbidities. Since ICU booking is predetermined, ICU admissions (yes/no) in our study are therefore independent of transfusion type, and this should be considered when interpreting the results. However, the number of days in the ICU (ICU LOS) was a valuable result. The RBCs&ICS group likely represented patients where (1) blood loss occurred suddenly (or unexpectedly) before ICS was available or (2) where lost blood volume could not be captured within the ICS process. Multitrauma surgery would be such an example, where patients experienced blood loss and received pRBCs before arrival at the theatre where ICS would be available. It is not possible to consider all potential confounding factors. The association between adverse outcomes and ABT, independent of confounding factors, has been confirmed by many investigators over the past 30 years.

Despite the retrospective nature of this study, important clinical measurements routinely collected during standard patient care were assessed, and missing data were relatively uncommon. This study included the collection of current relevant observational data with a large overall sample size (n = 2129), considered relevant confounding factors and patient characteristics, reported objectively recorded ICD-10 coded outcomes within an established exemplar ICS service, with no changes in protocol during the study period, used multivariable logistic regression models, and involved collaboration between experts in transfusion, information technology, database design, statistics, health economics, autotransfusion experts, and clinical anaesthetists.

The study design involved a retrospective observational analysis of nonrandomised data from a single centre, with the potential to be translated to a large patient population internationally. When considering the challenging nature and fast pace of many urgent procedures, inherent heterogeneity in both patient and clinical characteristics, ethical considerations, and the inability to truly blind ICS equipment, it is unlikely that an RCT design with traditional comparisons in these surgical procedures would soon be realistic. Specific cost values associated with individual adverse outcomes were not available; instead (as potentially important determinants of overall expenditure), LOS, ICU LOS, associated ICD-10 coded adverse outcomes, and blood product requirements were measured.

Future research directions should be considered. The use of database data in PBM will be essential in future to inform health policy, healthcare delivery, and implementation reform. Specific and selective biomarkers (modulated by transfusion) should be applied to future clinical outcome studies. A multicentre international trial including all potential confounding factors and surgical complexity, and powered to identify rare events, is needed to provide definitive answers. This manuscript provides the justification for such a future trial.

## 5. Conclusions

This study confirmed that allogeneic red blood cell transfusion was associated with significant downstream adverse outcomes and related costs. Previously confirmed ICS benefits (i.e., clinical outcomes, RBCs requirement, adverse outcomes, and ICU LOS) may extend beyond specific traditional surgical procedures (i.e., elective hip and knee replacement surgery). This study provides exploratory evidence and insights from real-world healthcare to assess the relevance and feasibility of future research into the hidden cost-saving potential of ICS. When implementing ICS, hospitals should not only consider direct costs (i.e., consumables, equipment, and staffing) but also consider the potential impact related to costs associated with adverse outcomes. However, this study does not provide the ultimate answer to this question. This evidence supports further investigation and does not dictate immediate clinical implementation. Large observational studies can provide valuable information to drive future multicentre research.

## Figures and Tables

**Table 1 healthcare-13-00503-t001:** Sources and examples of data collection: databases, comorbidity categories, and adverse outcome categories.

**Database Name (Abbreviated)**		**Example**
1	Operating Room Medical Information System (ORMIS)	Surgical procedure, surgical subspecialty
2	Integrated Electronic Medical Record (ieMR)	Demographic (age, gender, BMI)
3	Automated Anaesthetic Record Keeping (AARK)	Data cleaning (BMI, procedure)
4	Hospital-based corporate information system (HBCIS)	Length of stay in hospital (LOS)
5	Transfusion (Queensland Pathology)	Transfused time, number of units
6	Autotransfusion (Intraoperative Cell Salvage)	ICS processed or transfused
7	Health Information System (HIS)	ICD-10 coded adverse outcomes
8	Intensive Care Information System (Metavision)	Length of stay in intensive care (ICU LOS)
**Comorbidity Category**		**Example**
1	Cardiac disease	Ischaemic heart disease, cardiac failure
2	Cerebrovascular incidents	Ischaemic, haemorrhagic stroke and TIA
3	Diabetes mellitus	IDDM and NIDDM
4	Renal disease	Polycystic kidney disease, renal failure
5	Hypertension	Essential hypertension
6	Hypercholesterolaemia	
7	Respiratory disease	Asthma, COAD
8	Cancer	
9	Haematological	Sickle cell disease
**Adverse Outcome Category**		**Example**
1	Infection	Surgical wound infection
2	Respiratory	COAD
3	Renal disease	Acute renal failure
4	Cardiovascular	Acute myocardial infarction
5	Cerebrovascular	Stroke
6	Transfusion-related	ABO incompatibility
7	Medication	Drug reaction
8	Thromboembolism	Pulmonary embolism
9	Anaemia	

BMI, body mass index; ICS, intraoperative cell salvage; ICD-10, International Classification of Diseases-10; TIA, transient ischemic attack; IDDM, insulin-dependent diabetes mellitus; COAD, chronic obstructive airway disease; ABO, ABO-blood groups. Hospital data validation in Australia, following the “National Safety and Quality Health Service (NSQHS) Standards”, which are part of the Australian Health Service Safety and Quality Accreditation (AHSSQA) Scheme. This study only used data that were verified and included within hospital databases at the RBWH and followed NSQHS standards.

**Table 2 healthcare-13-00503-t002:** Transfusion incidence (%): comparison across subspecialties and study groups.

	RBCs (n = 1944)	ICS (n = 115)	RBCs&ICS (n = 70)	Overall (n = 2129)
Surgical Subspecialty	n (%)	n (%)	n (%)	n (%)
Orthopaedic	408 (21)	32 (28)	28 (40)	468 (22)
Obstetrics and Gynaecology	396 (20)	19 (17)	12 (17)	427 (20)
General	355 (18)	4 (3)	4 (6)	363 (17)
Vascular	206 (11)	55 (48)	23 (33)	284 (13)
Gastrointestinal	111 (6)	0	0	111 (5)
Plastic	110 (6)	0	0	110 (5)
Neurosurgery	102 (5)	3 (3)	1 (1)	106 (5)
Urology	81 (4)	2 (2)	0	83(4)
Burns	63 (3)	0	2 (3)	65 (3)
Ear, Nose and Throat	47 (2)	0	0	47 (2)
Maxillo-facial	34 (2)	0	0	34 (2)
Radiation oncology	15 (1)	0	0	15 (1)
Thoracic	12 (1)	0	0	12 (1)
Eyes	4 (0)	0	0	4 (0)

The subspecialties and procedures where ICS (intraoperative cell salvage) was not used indicated the potential “missed ICS opportunity”. Values are the number of patients (n) (proportion). pRBCs, packed red blood cells; ICS, intraoperative cell salvage; RBCs&ICS, red blood cells and intraoperative cell salvage.

**Table 3 healthcare-13-00503-t003:** Baseline characteristics of patients compared across study groups.

	RBCs (n = 1944)	ICS (n = 115)	RBCs&ICS (n = 70)		Overall (n = 2129)
Characteristics	Median (IQR)	Median (IQR)	Median (IQR)	*p* Value	Median (IQR)
Age in years	58 (38–71 [14–98])	61 (46–73 [15–85])	60 (35–72 [14–89])	0.36	58 (38–71 [14–98])
BMI	26.8 (23.0–31.6 [14.7–84.5])	27.5 (24.7–32.5 [16.9–61.0])	26.2 (22.9–30.6 [13.0–50.2])	0.16	26.8 (23.1–31.6 [13.0–84.5])
	**n (%)**	**n (%)**	**n (%)**		**n (%)**
Sex (n)					
Female	1055 (54.3%)	44 (38.3%)	41 (58.6%)	0.003	1140 (53.5%)
Male	889 (45.7%)	71 (61.7%)	29 (41.4%)		989 (46.5%)
Comorbidity Category					
0	559 (28.8%)	34 (29.6%)	22 (31.4%)	0.88	615 (28.9%)
1	375 (19.3%)	22 (19.1%)	15 (21.4%)		412 (19.4%)
2	286 (14.7%)	17 (14.8%)	13 (18.6%)		316 (14.8%)
3+	724 (37.2%)	42 (36.5%)	20 (28.6%)		786 (36.9%)
Adverse Outcome Category					
0	167 (8.6%)	42 (36.5%)	6 (8.6%)	<0.001	215 (10.1%)
1	931 (47.9%)	25 (21.7%)	27 (38.6%)		983 (46.2%)
2	385 (19.8%)	29 (25.2%)	11 (15.7%)		425 (20.0%)
3+	461 (23.7%)	19 (16.5%)	26 (37.1%)		506 (23.8%)

Those receiving RBCs (allogeneic red blood cells only), ICS (intraoperative cell salvage only), or RBCs&ICS (allogeneic red blood cells and intraoperative cell salvage) were compared. Values are median (IQR), n (number), and proportion (%). pRBCs, packed red blood cells; ICS, intraoperative cell salvage; RBCs&ICS, allogeneic red blood cells and intraoperative cell salvage; BMI, body mass index.

**Table 4 healthcare-13-00503-t004:** Results of length of stay, considering study groups and adjustment.

				LOS (Days)		LOS Cost (AUD)	
		n		Exp (β (95% CI))	Marginal Mean (95% CI)	Marginal Mean (95% CI)	*p* Value
Unadjusted							
Overall		2129					
RBCs		1944		Ref	18.4 (17.3–19.5)	39,321 (37,096–41,680)	
ICS		115		1.0 (0.8–1.2)	17.7 (13.9–22.5)	37,920 (29,843–48,184)	0.77
RBCs&ICS		70		1.2 (0.9–1.7)	22.0 (16.2–29.9)	47,159 (34,692–64,106)	0.25
Adjusted							
RBCs		1944		Ref	18.5 (17.1–20.0)	39,604 (36,583–42,873)	
ICS		115		1.0 (0.8–1.3)	18.4 (14.3–23.7)	39,467 (30,640–50,837)	0.98
RBCs&ICS		70		1.3 (0.9–1.8)	23.3 (16.6–32.7)	49,849 (35,530–69,935)	0.19
Age		2129		1.0 (1.0–1.0)	20.0 (17.3–23.0)	42,711 (36,992–49,315)	0.17
BMI		1776		1.0 (1.0–1.0)	20.0 (17.3–23.0)	42,711 (36,992–49,315)	0.93
Sex							
Female		1140		Ref	17.3 (14.7–20.2)	36,963 (31,568–43,280)	
Male		989		1.3 (1.2–1.5)	23.1 (19.7–27.0)	49,353 (42,177–57,750)	<0.001
Comorbidities Category							
0		615		Ref	16.1 (13.4–19.5)	34,567 (28,594–41,788)	
1		412		1.3 (1.1–1.6)	20.8 (17.3–25.2)	44,635 (36,945–53,925)	0.008
2		316		1.3 (1.0–1.6)	20.3 (16.6–25.0)	43,560 (35,514–53,430)	0.041
3+		786		1.4 (1.2–1.7)	23.1 (19.4–27.6)	49,514 (41,434–59,170)	<0.001
Adverse Outcomes Category							
0		215		Ref	14.9 (11.9–18.6)	31,803 (25,423–39,784)	
1		983		1.2 (0.9–1.5)	17.7 (15.0–20.9)	37,920 (32,153–44,720)	0.12
2		425		1.8 (1.4–2.3)	26.4 (21.9–31.9)	56,602 (46,885–68,333)	<0.001
3+		506		1.5 (1.2–1.9)	22.8 (19.0–27.3)	48,752 (40,686–58,418)	<0.001

Generalised linear model with gamma family and log link function. Values are β coefficient, exponentiated (exp) β coefficient (95% CI), marginal mean LOS in days (95% CI), and associated marginal mean LOS cost in AUD (95% CI). RBCs, allogeneic red blood cells; ICS, intraoperative cell salvage; RBCs&ICS, allogeneic red blood cells and intraoperative cell salvage; Exp, exponentiated; BMI, body mass index; Ref, reference.

**Table 5 healthcare-13-00503-t005:** Results of ICU admissions, considering study groups and adjustment.

	ICU Admission (y/n)			ICU LOS (Days)			Cost of ICU LOS (AUD) †
	n	Exp (β (95% CI))	*p* Value	Exp (β (95% CI))	*p* Value	Marginal Mean	Marginal Mean
Unadjusted							
Overall	2129						
RBCs	1944	Ref		Ref		6.4 (5.8–7.0)	29,958 (27,180–33,019)
ICS	115	2.4 (1.6-3.6)	<0.001	0.3 (0.2–0.4)	<0.001	2.0 (1.4–2.7)	9383 (6805–12,937)
RBCs&ICS	70	3.4 (2-5.7)	<0.001	0.7 (0.5–1.0)	0.044	4.2 (2.9–6.2)	19,853 (13,470–29,261)
Adjusted							
RBCs	1944	Ref		Ref		5.5 (4.8–6.3)	26,071 (22,770–29,850)
ICS	115	2.6 (1.7–4.1)	<0.001	0.4 (0.3–0.6)	<0.001	2.1 (1.5–2.9)	10,027 (7250–13,866)
RBCs&ICS	70	4.3 (2.4–8.0)	<0.001	0.7 (0.5–1.1)	0.081	3.8 (2.6–5.7)	18,089 (12,099–27,045)
Age	2129	1.0 (1.0–1.0)	0.61	1.0 (1.0–1.0)	0.001	3.6 (3.0–4.3)	16,785 (13,961–20,179)
BMI	1776	1.0 (1.0–1.0)	0.019	1.0 (1.0–1.0)	0.092	3.6 (3.0–4.3)	16,785 (13,961–20,179)
Sex							
Female	1140	Ref		Ref		3.2 (2.6–4.0)	15,036 (12,143–18,619)
Male	989	1.8 (1.5–2.2)	<0.001	1.2 (1.0–1.5)	0.025	4.0 (3.3–4.9)	18,736 (15,316–22,919)
Comorbidity Category							
0	615	Ref		Ref		3.8 (2.8–5.0)	17,721 (13,261–23,683)
1	412	1.2 (0.9–1.7)	0.21	1.1 (0.8–1.6)	0.44	4.3 (3.3–5.6)	20,097 (15,378–26,264)
2	316	1.6 (1.1–2.4)	0.007	0.8 (0.6–1.2)	0.31	3.2 (2.4–4.2)	14,860 (11,260–19,612)
3+	786	1.7 (1.2–2.5)	0.002	0.8 (0.6–1.2)	0.32	3.2 (2.6–4.0)	14,996 (12,039–18,679)
Adverse Outcome Category							
0	215	Ref		Ref		2.2 (1.5–3.2)	10,400 (7145–15,139)
1	983	1.4 (0.9–2.0)	0.097	1.4 (0.9–2.1)	0.12	3.0 (2.4–3.8)	14,355 (11,384–18,102)
2	425	4.6 (3.1–7.0)	<0.001	2.3 (1.5–3.4)	<0.001	5.1 (4.0–6.4)	23,854 (18,879–30,141)
3+	506	3.2 (2.2–4.9)	<0.001	2.1 (1.4–3.2)	<0.001	4.7 (3.8–5.9)	22,285 (17,711–28,041)

Results following the two-part hurdle model where ICU admission is modelled using binary logistic regression and † ICU LOS and cost is modelled using generalised linear models with gamma family and log link functions. Values are β, exponentiated (exp) β (95% CI), marginal mean ICU LOS in days (95% CI), and related marginal mean cost in AUD (95% CI). ICU, intensive care unit; LOS, length of stay; AUD, Australian dollars; RBCs, allogeneic red blood cells; ICS, intraoperative cell salvage; RBCs&ICS, allogeneic red blood cells and intraoperative cell salvage; Exp, exponentiated; BMI, body mass index; Ref, reference.

**Table 6 healthcare-13-00503-t006:** Blood product requirements and related costs.

		RBCs (n = 1944)		ICS (n = 115)		RBCs&ICS (n = 70)			Overall (n = 2129)	
		Mean mL (Range)		Mean mL (Range) (Range)		Mean mL (Range)			Mean mL (Range)	
ICS processed		1 (0–1156)		475 (0–3954)		536 (0–2721)			44 (0–3954)	
ICS transfused		NA		413 (78–1545)		472 (31–2700)			38 (0–2700)	
**Allogeneic blood products**	**Unit cost * (AUD/unit)**	**Number of units (range)**	**Total cost (cost per patient) (AUD)**	**Number of units (range)**	**Total cost (cost per patient) (AUD)**	**Number of units (range)**	**Total cost (cost per patient) (AUD)**	***p* Value**	**Number of units (range)**	**Total cost (cost per patient) (AUD)**
Aph platelets	634	327 (0–30)	207,387 (107)	2 (0–2)	1268 (11)	6 (0–2)	3805 (54)	0.055	335 (0–30)	212,460 (100)
Pooled Platelets	278	708 (0–31)	197,008 (101)	1 (0–1)	278 (2)	20 (0–4)	5565 (80)	0.002	729 (0–31)	202,852 (95)
Plasma	182	1061 (0–44)	192,879 (99)	2 (0–2)	364 (3)	79 (0–20)	14,361 (205)	<0.001	1142 (0–44)	207,604 (98)
Aph plasma	264	60 (0-8)	15,811 (8)	0	0	0	0	0.36	60 (0–8)	15,811 (7)
Cryo	164	1190 (0–45)	195,160 (100)	1 (0–1)	164 (1)	76 (0–20)	12,464 (178)	0.008	1267 (0–45)	207,788 (98)
Aph cryo	320	717 (0–25)	229,727 (118)	11 (0–6)	3524 (31)	66 (0–10)	21,146 (302)	0.002	794 (0–25)	254,398 (119)
pRBCs	399	8884 (1–99)	354,6937 (1825)	NA		361 (1–29)	144,129 (2059)	0.008	9245 (0–99)	3,691,066 (1734)
Total cost (Including pRBCs)			4,584,909 (2358)		5599 (48)		201,471 (2878)	<0.001		4,791,979 (2251)
Total cost (Excluding pRBCs)			1,037,972 (533)		5599 (48)		57,342 (819)			1,100,913 (517)

RBCs, allogeneic red blood cells; ICS, intraoperative cell salvage; RBCs&ICS, allogeneic red blood cells and intraoperative cell salvage; values are for n number of patients; mL, millilitre; ICS, intraoperative cell salvage volume in mL (range); Aph, apheresis; Cryo, cryoprecipitate; RBCs, allogeneic red blood cells and other products (unit cost, adjusted to 2020 *), number of units (range), total cost for specific blood product type (average cost per patient for specific blood product type). All costs are in AUD, Australian dollars.

**Table 7 healthcare-13-00503-t007:** Adverse outcomes are summarised within organ-specific categories.

	RBCs		ICS		RBCs&ICS		Overall		
	(n = 1944 (a))		(n = 115 (b))		(n = 70 (c))		(n = 2129 (d))		
Number of Adverse Outcomes	n *	Median (IQR)	n *	Median (IQR)	n *	Median (IQR)	n *	Median (IQR)	*p* Value
	3689	1 (1–2 [0–15])	181	1 (0–2 [0–15])	159	2 (1–3 [0–7])	4029	1 (1–2 [0–15])	<0.001
**Patients with Adverse Outcomes**	** n ** (% of a)**		** n ** (% of b)**		** n ** (% of c)**		** n ** (% of d)**		
Overall	1777 (91.4)		73 (63.5)		64 (91.4)		1914 (89.9)		<0.001
Infection	476 (24.5)		19 (16.5)		14 (20.0)		509 (23.9)		0.11
Respiratory	388 (20.0)		29 (25.2)		22 (31.4)		439 (20.6)		0.030
Renal	204 (10.5)		8 (7.0)		7 (10.0)		219 (10.3)		0.48
Cardiovascular	934 (48.0)		44 (38.3)		46 (65.7)		1024 (48.1)		0.001
Cerebrovascular	20 (0.0)		1 (0.9)		0		21 (1.0)		1.00
Medication related	40 (2.1)		0 (0)		1 (1.4)		41 (1.9)		0.35
Thromboembolism	41 (2.1)		3 (2.6)		3 (4.3)		47 (2.2)		0.31
Anaemia	765 (39.4)		16 (13.9)		24 (34.3)		805 (37.8)		<0.001

RBCs, allogeneic red blood cells; ICS, intraoperative cell salvage; RBCs&ICS, allogeneic red blood cells and intraoperative cell salvage. Values are median (IQR) or number (proportion); n number of patients per group; n * number of adverse outcomes; n ** number of patients with adverse outcomes.

## Data Availability

The dataset supporting the conclusions of this article is included within the manuscript in a deidentified format.

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
