# Peer review of "The Cost of Downstream Adverse Outcomes Associated with Allogeneic Blood Transfusion: A Retrospective Observational Cohort Study"

_healthcare, 2025, doi:10.3390/healthcare13050503_

Round 1
Reviewer 1 Report
Comments and Suggestions for Authors
The Cost of Downstream Adverse Outcomes Associated with 2 Allogeneic Blood Transfusion: a retrospective observational cohort study.
Review
This is so important subject, with high priority specially in practice, but this important procedure needs high level equipment hospital which may be not accessible in low income countries.
|
Lines |
Present |
Notes |
|
59 |
Many references |
|
|
105 |
No references |
Need to be reviewed |
|
107 |
Material and method |
Patients and methods |
|
107 |
Material and methods |
Many details which may be can summarized |
|
305 |
ICS is not standardized across countries and hospitals |
It is important points, which may reflected on results of this paper and discussion |
|
311
|
The value of multicenter comparisons may therefore be questioned |
Multicenter study comparison is essential point to give this novel subject real data |
|
469 |
Conclusion |
It need to be summarize, more details may hide the real idea of paper |
Author Response
References to line numbers relate to the revised manuscript. Please find replies to reviewer comments (in blue) below in black:
“This is so important subject, with high priority specially in practice, but this important procedure needs high level equipment hospital which may be not accessible in low income countries”.
Dear Reviewer 1, Thank you. Indeed, this is an important subject.
Absolutely, this is the purpose of this manuscript. The cost of the equipment is often the reason why cell savage (ICS) is not implemented. We therefore did this study to show the costs associated with the alternative, allogeneic blood transfusion. Because the downstream costs are often hidden and difficult to study many are not familiar with these cost consequences.
In my opinion modern ICS technology should be simplified. More work and research should go into developing simplified ICS technology. By publishing this manuscript, I hope we can highlight this very important fact and encourage researchers and industry to develop technology that would be affordable and simple, so that it can indeed be used in all countries.
Line 59
Dear Reviewer, may we please ask clarification on the question in this point?
In line 105: “no references, Need to be reviewed”
This relates to the objectives in our study. Please find reference now included line 119 of the revised manuscript: “Supplemental table 1”.
Line 107 is blank. On line 108, Re “Material and methods”,
We believe “Material and methods” is the heading recommended by the Healthcare journal? We would accept any changes to this heading by the editor, if required. Thank you.
Line 108, “Material and methods”, “Many details which may be can summarized“
It is unclear what is meant by “Many details which may be can summarized“. Does reviewer recommend that we summarise the full method section? The other 2 reviewers recommended adding more information to this section. Please see revised manuscript.
Line 305: Re “ICS is not standardized across countries and hospitals“
“It is important points, which may reflected on results of this paper and discussion”
Thank you. Indeed, we agree. Please refer to lines 347-354: “ICS is not standardized across countries and hospitals” with the consequences and considerations of this relevant to this manuscript in the discussion section.
Because of this, it would be challenging to consider those differences across countries and hospitals in a future trial. This is one of the reasons why I believe as a starting point (to explore opportunities) using one centre (as in our study) does overcome the controversies related to this point. ICS is standardised in our hospital.
Please find added to discussion in line 355 of the revised manuscript: “Future multicenter international trials that consider these potential benefits are needed. However, the design of such studies that will overcome the challenges related to institutional differences in the ICS procedure will be complicated.”
Line 311: Re “Multicenter study comparison is essential point to give this novel subject real data’
Thank you. See included as per above comment. We agree that a multicentre study will be required to discover the ultimate answer to this research question. In this instance due to the differences in technique and policies we started with a single centre study, so that we could at least keep these factors similar across study patients for the purposes of statistical design.
Line 469: Re “Conclusion”. “It need to be summarize, more details may hide the real idea of paper”
Please find summarised conclusion line 543: “This study confirmed that allogeneic red blood cell transfusion was associated with significant downstream adverse outcomes and related costs. Previously confirmed ICS benefits (i.e., clinical outcomes, RBC requirement, adverse outcomes, and ICU LOS) may extend beyond specific traditional surgical procedures (i.e., elective hip and knee replacement surgery). This study provide exploratory evidence and insights from real-world health care to assess the relevance and feasibility of future research into the hidden cost-saving potential of ICS. When implementing ICS, hospitals should not only consider direct costs (i.e., consumables, equipment and staffing) but also consider the potential impact related to costs associated with adverse outcomes. However, this study does not provide the ultimate answer to this question. This evidence supports further investigation and does not dictate immediate clinical implementation. Large observational studies can provide valuable information to drive future multicentre research.”
We moved other important considerations to the discussion section. Please find changes on line 490: “Future ICS research should not only concentrate on historical equipment and staffing costs but also consider costs avoided when adverse outcomes are prevented. Current evidence and expert opinion should guide the allocation of funding in these circumstances. ICS would never be relevant for all surgical procedures, but resource allocation toward its wider implementation would be justifiable.”
And line 528: “When considering the challenging nature and fast pace of many urgent procedures, inherent heterogeneity in both patient and clinical characteristics, ethical considerations and the inability to truly blind ICS equipment, it is unlikely that an RCT design with traditional comparisons in these surgical procedures would soon be realistic.”
Reviewer 2 Report
Comments and Suggestions for Authors
Minor Comments
1. In the fourth paragraph of the introduction (page 2 lines 77–79), it would be beneficial to briefly mention studies focusing on transfusion-related immunomodulation (TRIM) outcomes. For example ,Li examined the impact of donor and recipient sex on sepsis rates and hemoglobin recovery in critically ill transfusion patients, finding that sex mismatches may influence TRIM-related complications and post-transfusion outcomes(Li et al. 2025). Additionally, Ning et al. discussed how big data analytics and ICD-10 coding improve research on TRIM and adverse outcomes like infections and organ dysfunction (Ning et al. 2022).
2. Page 4 line 126 Table 2 title can be revised. Table 2. To provide clinical context, we considered the transfusion incidence for each subspecialty across study group can be revised to Table 2. Transfusion Incidence (%): Comparison Across Subspecialties and Study Groups.
3. Page 5 table 1: The table is informative but lacks clarity on data validation produces.
4. Page 8 table 3: The statical difference on sex distraction across study groups may indicate a potential confounder affecting outcomes. consider adjusting for sex in multivariable models or commenting on its potential influence in the discussion section.
5. Page 10, line 234 -241: The cost differences between study groups are presented clearly, but the rationale for choosing a gamma family distribution for modeling LOS is missing. Briefly justify the use of the gamma distribution and log-link function, particularly in handling skewed data.
6. Page 17, line 337-341:The discussion acknowledges challenges in directly linking data across databases, but does not specify how potential errors were mitigated.
Uncategorized References
Li, W., Y. Liu, K. J. Lucier, N. M. Heddle, and J. P. Acker. 2025. 'The association of donor and recipient sex on sepsis rates and hemoglobin increment among critically ill patients receiving red cell transfusions in a retrospective study', EJHaem, 6: e1005.
Ning, Shuoyan, Na Li, Rebecca Barty, Donald Arnold, and Nancy M. Heddle. 2022. 'Database-driven research and big data analytic approaches in transfusion medicine', Transfusion, 62: 1427-34.
Author Response
References to line numbers relate to the revised manuscript. Please find replies to reviewer comments (in blue) below in black:
Minor Comments
In the fourth paragraph of the introduction (page 2 lines 77–79), it would be beneficial to briefly mention studies focusing on transfusion-related immunomodulation (TRIM) outcomes. For example ,Li examined the impact of donor and recipient sex on sepsis rates and hemoglobin recovery in critically ill transfusion patients, finding that sex mismatches may influence TRIM-related complications and post-transfusion outcomes(Li et al. 2025). Additionally, Ning et al. discussed how big data analytics and ICD-10 coding improve research on TRIM and adverse outcomes like infections and organ dysfunction (Ning et al. 2022).
Dear Reviewer, 2, Thank you. We agree with Li et al, there are potentially many associated confounding factors and outcomes hidden within the patient journey. I will reference Li et al as well as my PhD thesis [15]; recently conferred with the University of Queensland: “Roets M. PhD thesis. Intraoperative cell salvage as an alternative to allogeneic blood transfusion: an evaluation of immune related adverse outcomes, Faculty of Medicine. UQ eSpace, The University of Queensland, Faculty of Medicine, 2024, pp 1-257”.
Cited [20]: Li, W., Y. Liu, K. J. Lucier, N. M. Heddle, and J. P. Acker. 2025. 'The association of donor and recipient sex on sepsis rates and hemoglobin increment among critically ill patients receiving red cell transfusions in a retrospective study', EJHaem, 6: e1005.
We agree with Ning et al. and will include the reference. Thank you. Database analysis, in future, will allow inclusion of all the confounding factors that we are currently not able to consider and will subsequently shed light on hidden factors. It is not possible to include all relevant confounders in a study such as ours, due to the data not being available in existing databases as well as the inability to specifically connect some of the data in various databases to specific time points in the patient journey. These confounding factors are noted as limitations of our study (lines 496-516).
Furthermore, we agree with Ning et al. that ICD-10 is conservative (“limited sensitivity (62.1–99.7%) and specificity (78.3–100%)”. The outcomes we considered reflect the tip of the iceberg. The reason we used ICD-10 coding is however that it is objective and removes some investigator bias. The incidence of TRIM related adverse outcomes may be much larger than identified in our manuscript. However, we did want to initiate this concept, that the clinical impact of TRIM may involve other outcomes and procedures, in addition to those traditional outcomes (i.e., wound infection and cancer recurrence).
Please find included on line 92:” We agree with Ning et al. who in a narrative review considers the value and challenges relevant to “big data” in transfusion medicine in future. There is a potential that database analysis and artificial intelligence will in future lead to improved clarity of the relevance of outcomes related to TRIM and associated organ dysfunction5”
Cited [23]: Ning, Shuoyan, Na Li, Rebecca Barty, Donald Arnold, and Nancy M. Heddle. 2022. 'Database-driven research and big data analytic approaches in transfusion medicine', Transfusion, 62: 1427-34.
- Page 4 line 126 Table 2 title can be revised. Table 2. To provide clinical context, we considered the transfusion incidence for each subspecialty across study group can be revised to Table 2. Transfusion Incidence (%): Comparison Across Subspecialties and Study Groups.
Thank you. Please find the following change on line 248: “Transfusion Incidence (%): Comparison Across Subspecialties and Study Groups”
- 3. Page 5 table 1: The table is informative but lacks clarity on data validation produces.
Thank you. Please find added to the revised manuscript on line 151: “*Hospital data validation in Australia follow the "National Safety and Quality Health Service (NSQHS) Standards", which are part of the Australian Health Service Safety and Quality Accreditation (AHSSQA) Scheme. This study only used data that were verified and included within hospital databases at the RBWH and followed NSQHS standards.
- 4. Page 8 table 3: The statical difference on sex distraction across study groups may indicate a potential confounder affecting outcomes. consider adjusting for sex in multivariable models or commenting on its potential influence in the discussion section.
Thank you. We agree. Sex is a potentially confounding factor. Please refer to line 240 in the statistical analysis section: “All statistical models presented were run with the treatment group alone as well as with adjustment for the potential known confounding factors of age, sex, BMI, preexisting comorbidities, and adverse outcomes”
- 5. Page 10, line 234 -241: The cost differences between study groups are presented clearly, but the rationale for choosing a gamma family distribution for modeling LOS is missing. Briefly justify the use of the gamma distribution and log-link function, particularly in handling skewed data.
Thank you. Please find now included, in the revised manuscript on line 231: “Study design and statistical methods were chosen following extensive consultation with an experienced biostatistician and health economist from the start of the study to manuscript preparation. A generalized linear model with a gamma distribution and log link (commonly used to model costs and length of stay in health care data) was appropriate to be used when the outcome was positive, continuous and skewed. The gamma distribution could accommodate differing levels of skewness and provided a reasonably good fit when there was heteroscedasticity. The log link function allowed the modelling of multiplicative relationships (i.e. evaluate relative changes). It also reduced the impact of extreme values and stabilized variance.”
- 6. Page 17, line 337-341:The discussion acknowledges challenges in directly linking data across databases, but does not specify how potential errors were mitigated.
Thank you. Please find included on line 158, of the revised manuscript: “Databases were not linked due to compatibility issues and different software languages between software systems used in relevant databases. Instead, data was retrieved from eight hospital databases, exported onto an Excel spreadsheet and then imported into a specifically designed novel Structured Query Language (SQL) database developed to link data points. Import of data within three monthly cycles over a 2-year period allowed for extensive query analysis, to ensure potential errors were mitigated. Specific strategies were followed at each data import and linkage step (i.e., collect, connect and clean) involving professional information technology experts. In addition, each data import step was checked by five study members against original exported data. An additional check (data cleaning) was included within statistical analysis. Exporting data from existing hospital databases rather than using manual importing of data did minimize data entry errors. Extra checks were included for doubled data and outliers.”
Reviewer 3 Report
Comments and Suggestions for Authors
Roets et al submitted an interesting original article investigating the economic and clinical impact of allogeneic blood transfusion (ABT) compared to intraoperative cell salvage (ICS) in surgical patients. The study highlights significant cost differences and adverse outcome rates associated with transfusion modalities through using a retrospective cohort design with data from 2,129 patients. The findings are valuable for informing PBM strategies; however, several aspects need to take into account to improve the manuscript prior to the publication as follow:
1. Abstract: I would suggest including specific quantitative data (e.g., mean hospital costs, adverse outcome percentages). In addition to that, authors should clarify that findings are limited to retrospective data and require further prospective validation
2. Introduction: I would recommend to use more recent references on the discussion of transfusion-related immune modulation (TRIM). Authors should also provide additional details on why previous studies have not sufficiently addressed cost implications and specify hypotheses to clearly define the study’s objectives.
3. Methods: I would like to know the reason why adjustments were made for potential confounders such as preoperative hemoglobin levels and surgical complexity. Also, I will be happy to know the linkage across eight hospital databases to enhance transparency.
4. Results: I feel that some tables (e.g., cost analysis) are dense. I have also a concern about the mortality rates as they should be more explicitly discussed to determine whether differences in adverse outcomes translate to survival benefits.
5. Discussion: I would suggest that authors discuss more on potential biases, particularly regarding patient selection for ICS. Also, I would like to know more about cost-effectiveness rather than just cost comparisons. Future research directions should be included.
6. Conclusions: I would suggest avoiding overstating clinical implications; instead, emphasize that findings support further investigation rather than immediate clinical implementation. Also, policy recommendations or healthcare implications based on the data should be mentioned.
Author Response
References to line numbers relate to the revised manuscript. Please find replies to reviewer comments (in blue) below in black:
Roets et al submitted an interesting original article investigating the economic and clinical impact of allogeneic blood transfusion (ABT) compared to intraoperative cell salvage (ICS) in surgical patients. The study highlights significant cost differences and adverse outcome rates associated with transfusion modalities through using a retrospective cohort design with data from 2,129 patients. The findings are valuable for informing PBM strategies; however, several aspects need to take into account to improve the manuscript prior to the publication as follow:
Dear Reviewer 3, Thank you for the positive comments. We agree that this study does not provide the ultimate answer. The main aim was to highlight the fact that patients who receive ABT (for various reasons that are outside the limits of this study and demonstrated throughout the literature) do still experience adverse outcomes, that blood products are expensive and that these patients require extensive intensive care and costs.
We also know from many previous studies that ICS potentially provides benefits and improved outcomes. Yet an ongoing and common discussion (in the literature and hospital business meetings), is that ICS consumables, staffing and equipment are expensive. We therefore wanted to highlight the fact that the alternative may be more expensive and that there may be a great avenue for investigation, and potentially improvement, when considering costs and that future investigation and research into ICS technology may therefore be of value.
- Abstract: I would suggest including specific quantitative data (e.g., mean hospital costs, adverse outcome percentages). In addition to that, authors should clarify that findings are limited to retrospective data and require further prospective validation
Specific hospital costs associated with these specific TRIM related outcomes (not commonly studied) considered in this manuscript (wider inclusion than traditional infection and cancer recurrence outcomes, 171 additional ICD-10 coded outcomes) are not available. Added to line 198 find “171”. Alternative determinants of overall expenditure were therefore assessed.
Please refer to line 210: “As potentially important determinants of overall expenditure LOS, ICU LOS, associated ICD-10 coded adverse outcomes, blood product requirements and related costs”.
Considering mean hospital costs: As “surrogate” markers of the fact that hospitals incur costs we used LOS and ICU LOS costs. Previously defined (and used cost indicators across Australia) aggregated costs for LOS (AU$2,140.81 per day)[27] and ICU LOS (AU$4,717.32 per day)[28] were applied.
Please find included on line 42: “As potentially important determinants of overall expenditure,”
Please refer to line 43 “The lowest marginal mean intensive care stay (days, cost) was after ICS (2.1 days, $10,027), followed by RBC and ICS (3.8 days, $18,089) and RBC (5.5 days, $26,071).”
Considering adverse outcome percentages and quantitative data (“adverse outcome percentages”), please refer to line 41: “Most adverse outcomes (≥3) occurred following RBC and ICS (37.1%), followed by RBC (23.7%) and ICS (16.5%).”
Please find now included on line 49 of the revised manuscript: “Results are limited to retrospective data and require further prospective validation”. Thank you.
- Introduction: I would recommend to use more recent references on the discussion of transfusion-related immune modulation (TRIM). Authors should also provide additional details on why previous studies have not sufficiently addressed cost implications and specify hypotheses to clearly define the study’s objectives.
Considering more recent references:
Thank you. Please find referenced Roets et al 2019 [1] line 56 and
Roets et al 2024. [15] on line 72. This is my PhD thesis published in 2024; recently conferred with the University of Queensland: “Roets M. PhD thesis. Intraoperative cell salvage as an alternative to allogeneic blood transfusion: an evaluation of immune related adverse outcomes, Faculty of Medicine. UQ eSpace, The University of Queensland, Faculty of Medicine, 2024, pp 1-257”.
Also, please find references included (as per reviewer 2):
Li, W., Y. Liu, K. J. Lucier, N. M. Heddle, and J. P. Acker. 2025. 'The association of donor and recipient sex on sepsis rates and hemoglobin increment among critically ill patients receiving red cell transfusions in a retrospective study', EJHaem, 6: e1005.
And
Ning, Shuoyan, Na Li, Rebecca Barty, Donald Arnold, and Nancy M. Heddle. 2022. 'Database-driven research and big data analytic approaches in transfusion medicine', Transfusion, 62: 1427-34.
Considering “details on why previous studies have not sufficiently addressed cost implications”.
Thank you. Please find included on line 336: “The reasons why previous studies have not sufficiently addressed cost implications”.
Considering “specify hypotheses to clearly define the study’s objectives”
Thank you. Please find additional information, in the revised manuscript on lines 27-34: “The main aim of this retrospective observational study was to evaluate the cost implications associated with perioperative adverse outcomes following allogeneic blood transfusion (ABT). Secondly, further analysis considered downstream costs following ICS. This manuscript does not aim to provide evidence of improved outcomes following ICS, compared to ABT. These outcomes were previously demonstrated. Instead, it is important to consider downstream cost implications if patients receive ABT, despite previously proven benefits related to ICS. “
- Methods: I would like to know the reason why adjustments were made for potential confounders such as preoperative hemoglobin levels and surgical complexity. Also, I will be happy to know the linkage across eight hospital databases to enhance transparency.
Considering “reason why adjustments were made for potential confounders such as preoperative hemoglobin levels and surgical complexity”
Dear reviewer 3. Thank you. These are very important and valid comments.
Anaemia, preoperative haemoglobin and surgical complexity were not included as confounding factors. Anaemia as an ICD-10 code (as in our study), is considered an outcome with cost implications to hospitals.
Including surgical complexity (even though not possible from standard hospital database data in our hospital) would be an important confounding factor for future research.
However, these factors (anaemia and surgical complexity) were previously considered in many studies that demonstrated the advantages of ICS (i.e., studies that evaluated the clinical benefits of ICS vs ABT). We accepted those results (i.e., the fact that ICS is potentially beneficial). In other words, the aim of the study was not to repeat those studies (i.e., see whether patients do better if they receive ABT or ICS). Instead, we aimed to count the cost when patients received ABT and when they received ICS (i.e., considering proven benefits). We then recommend future prospective observational studies in large sample sizes that can answer remaining questions more definitively. This study does not aim to provide the ultimate answer, but instead to provide supporting evidence that ICS may actually be more cost beneficial if outcomes and costs are considered (i.e., another consideration to build into the background when discussing ICS cost, in addition to the traditional “consumable cost” evaluations).
Considering “linkage across eight hospital databases”, please find now included on line 158: “Databases were not linked due to compatibility challenges between software systems used in relevant databases. Instead, data retrieved from eight hospital databases, was exported onto excel and then imported into a Structured Query Language (SQL) database to link data points; with a query analysis in three monthly cycles over a 2-year period. Specific strategies were followed at each data import and linkage step (i.e., collect, connect and clean) involving information technology experts. In addition, each data import step was checked by five study members against original exported data. An additional check (data cleaning) was included within statistical analysis. Exporting data from existing hospital databases rather than using manual importing of data did minimize data entry errors. Extra checks were included for doubled data and outliers.”
- Results: I feel that some tables (e.g., cost analysis) are dense. I have also a concern about the mortality rates as they should be more explicitly discussed to determine whether differences in adverse outcomes translate to survival benefits.
Thank you. Indeed, there are so many controversies around “the use of ICS and cost”, that we did aim to describe our findings in as much detail as possible and with extensive consultation with an experienced biostatistician and health economist, leaving us with very detailed tables.
When considering mortality rates (an important consideration), this explorative study was insufficiently powered (small sample sizes) to consider mortality rates (ICS only (n=115), allogeneic red blood cells (RBC) only (n=1,944) or RBC and ICS (n=70)). We therefore did not do a statistical analysis considering mortality as it was clear from our data and previous literature reviews that did consider mortality rates in transfusion research, that our study would not be sufficiently powered to evaluate mortality as an outcome.
Please find on line 323 of the revised manuscript: “Of the 2,129 patients in our study, 82 died in the hospital (3.9%).”
Refer to line 84: Instead, the study design focussed on exploration of outcomes (morbidity) and procedures not commonly included in traditional ICS research.
- 5. Discussion: I would suggest that authors discuss more on potential biases, particularly regarding patient selection for ICS. Also, I would like to know more about cost-effectiveness rather than just cost comparisons. Future research directions should be included.
Considering “potential biases”:
Thank you. This is very important. There are however no specific patient selection criteria beyond those mentioned in the manuscript.
The main aim of the study was to consider cost associated with allogeneic transfusion. And then secondly to consider the cost associated with ICS for the current cohort. ICS is available at any time of the day, any day of the year at the RBWH. We do not use ICS only for a list of predefined procedures.
Please find previously included and on line 128 of the revised manuscript: “ICS is booked at the RBWH at the discretion of the surgeon and anaesthetist when major blood loss (i.e., more than 500 mL) is expected, for any procedure. ICS is available within and outside office hours, for elective and emergency cases, every day of the year at the RBWH (and not only for a list of predefined procedures). There are no absolute contraindications to ICS”
Please refer to lines 496-515: “There were some limitations to our study, including its retrospective data collection, single center design and observational nature.”
Considering cost-effectiveness:
Sample sizes considering ICS in a single centre does unfortunately not allow sufficient power to consider cost-effectiveness.
An extensive cost-effectiveness analysis considering traditional cost implications is available by Davis et al (reference [18]). These authors recommended observational studies to explore considerations that were previously not included in studies considering an RCT design. Our study (Roets et al) was designed specifically considering recommendations from Davies et al., following advice from this cost-effectiveness analysis (Davies et al).
Please refer to line 86: “We could however not find any studies that included all transfusion-related adverse outcomes associated with immune modulation (TRIM) (e.g., not only wound infection but also pneumonia, renal failure, cardiovascular and stroke-related outcomes, etc.), used objective and standardised ICD-10 coding and included an assessment of the significance and impact of the associated cost burden (considering pRBCs and other blood products, length of stay in hospital (LOS) and in the intensive care unit (ICU)).”
Therefore, we aimed to answer specific knowledge gaps (considering and attempting to reduce previously mentioned limitations) in the literature (line 96). To start the conversation, we aim to provide explorative evidence to overcome obstacles to the growth of ICS (i.e., The traditional controversy that ICS is expensive due to consumables, equipment and staffing). Limited research considered the cost implications to hospitals considering morbidity following transfusion.
Considering “patient selection for ICS”.
Furthermore, many considered only specific surgical procedures. In view of the fact that there are no remaining absolute contraindications to ICS now, modern ICS research should consider all surgical procedures requiring ABT. May I debate that there should no longer be a “patient selection for ICS”. There are no remaining absolute contraindications to ICS (i.e., Klein A [13]), therefore we did not exclude any specific procedures considering ICS in this study.
Many advantages following ICS have been confirmed (Lloydla [2], Carless [14] etc.). We therefore aimed to assess the cost associated with adverse outcomes, for all surgical procedures requiring transfusion, where ICS may be of benefit.
Considering “Future research directions should be included”:
Please find included on line 535 of the revised manuscript “Future research directions should be considered. The use of database data in PBM will be essential in future to inform health policy, healthcare delivery and implementation reform. Specific and selective biomarkers (modulated by transfusion) should be applied to future clinical outcome studies. A multicenter international trial including all potential confounding factors and surgical complexity, and powered to identify rare events, is needed to provide definitive answers. This manuscript does provide the justification for such a future trial.”
- Conclusions: I would suggest avoiding overstating clinical implications; instead, emphasize that findings support further investigation rather than immediate clinical implementation. Also, policy recommendations or healthcare implications based on the data should be mentioned.
Please find changed conclusion on line 543 of the revised manuscript: “This study confirmed that allogeneic red blood cell transfusion was associated with significant downstream adverse outcomes and related costs. Previously confirmed ICS benefits (i.e., clinical outcomes, RBC requirement, adverse outcomes, and ICU LOS) may extend beyond specific traditional surgical procedures (i.e., elective hip and knee replacement surgery). This study provide exploratory evidence and insights from real-world health care to assess the relevance and feasibility of future research into the hidden cost-saving potential of ICS. When implementing ICS, hospitals should not only consider direct costs (i.e., consumables, equipment and staffing) but also consider the potential impact related to costs associated with adverse outcomes. However, this study does not provide the ultimate answer to this question. This evidence supports further investigation and does not dictate immediate clinical implementation. Large observational studies can provide valuable information to drive future multicentre research.”